# From Density Functional Theory to Conceptual Density Functional Theory and Biosystems

**DOI:** 10.3390/ph15091112

**Published:** 2022-09-06

**Authors:** Paul Geerlings

**Affiliations:** Research Group of General Chemistry (ALGC), Faculty of Science and Bio-Engineering Science, Vrije Universiteit Brussel (VUB), Pleinlaan 2, B-1050 Brussels, Belgium; pgeerlin@vub.be

**Keywords:** DFT, conceptual DFT, response functions, reactivity descriptors, CDFT principles, applications, enzymatic catalysis, biological activity, toxicity, computational peptidology

## Abstract

The position of conceptual density functional theory (CDFT) in the history of density functional theory (DFT) is sketched followed by a chronological report on the introduction of the various DFT descriptors such as the electronegativity, hardness, softness, Fukui function, local version of softness and hardness, dual descriptor, linear response function, and softness kernel. Through a perturbational approach they can all be characterized as response functions, reflecting the intrinsic reactivity of an atom or molecule upon perturbation by a different system, including recent extensions by external fields. Derived descriptors such as the electrophilicity or generalized philicity, derived from the nature of the energy vs. N behavior, complete this picture. These descriptors can be used as such or in the context of principles such as Sanderson’s electronegativity equalization principle, Pearson’s hard and soft acids and bases principle, the maximum hardness, and more recently, the minimum electrophilicity principle. CDFT has known an ever-growing use in various subdisciplines of chemistry: from organic to inorganic chemistry, from polymer to materials chemistry, and from catalysis to nanotechnology. The increasing size of the systems under study has been coped with thanks to methodological evolutions but also through the impressive evolution in software and hardware. In this flow, biosystems entered the application portfolio in the past twenty years with studies varying (among others) from enzymatic catalysis to biological activity and/or the toxicity of organic molecules and to computational peptidology. On the basis of this evolution, one can expect that “the best is yet to come”.

## 1. From DFT to Conceptual DFT

It is not an overstatement to say that density functional theory (DFT) revolutionized quantum chemistry and in particular its computational part, nowadays called computational chemistry. Today, DFT is the workhorse “par excellence” when exploring structure, stability, electronic properties, reactivity and reactions of molecules, polymers, and solids in the most diverse subdomains of chemistry, covering, thanks to the inclusion of relativistic effects, nearly the complete periodic table [1,2]. This revolution has been realized based on the ingenious idea of using the electron density ρ(**r**) instead of the wave function Ψ as the basic carrier of information. The simplification is spectacular: for an N-electron system, one switches from an immensely complicated wave function Ψ(**x**^N^), a function of 4N variables (three spatial and one spin variable for each electron, gathered in a four-vector **x**), to only three variables in the density ρ(x,y,z). Although the density concept has been present from the early days of quantum mechanics in, e.g., the Thomas Fermi *model* [3,4], the decisive step towards a full-fledged density functional *theory* was taken by Hohenberg and Kohn [5] through their two famous theorems. The first theorem, an existence theorem, proves that the ground state energy of a system E is a function of the density ρ(**r**). The second theorem introduces a variational principle, and thereby offers a road to the “best” density by searching for the one yielding the lowest energy, an ansatz known for decades in wave function quantum mechanics. The first theorem, for which the proof was quoted “disarmingly simple” by Parr [6] but which is particularly ingenious in its construction, proves by a reductio ad absurdum that the ground state density ρ(**r**) is compatible with a single external potential v(**r**), i.e., the potential felt by the electrons due to the nuclei, in the absence of external fields. This single external potential corresponds to a unique “constellation” of nuclei: their number, position, and charge. Stated otherwise: ρ determines v. As ρ also determines N by integration, it also determines the Hamiltonian and, at least in principle, “everything”. Through the variational procedure (the second theorem) the “best” ρ results from solving the Euler equation of the problem:v(**r**) + δF_HK_/δρ(**r**) = μ(1)
where F_HK_ is the Hohenberg Kohn functional and μ is the Lagrange multiplier introduced during the variational procedure, ensuring that the density remains properly normalized to N. This equation is the analogue of the time independent Schrödinger equation H Ψ = E Ψ, which can also be obtained in a variational ansatz. In this case, the Lagrange multiplier is introduced to ensure that the proper normalization of the wavefunction Ψ is at the end identified with the system’s energy E. 

Equation (1) deserves particular attention from the computational point of view but also, and in particular in this introductory paper, in view of the passage to conceptual DFT. What are, besides ρ and v, the two remaining ingredients, F_HK_ and μ, of this equation? F_HK_ is the Hohenberg Kohn functional, a universal (i.e., v independent) functional containing unknown parts governing electron correlation and exchange gathered in the exchange-correlation functional Exc [ρ]. To cope with this issue, an ingenious step was taken by Kohn and Sham [7]: the introduction of orbitals, in the context of a non-interacting reference system. They thereby succeeded in transforming the variational equation into a series of pseudo one-electron eigenvalue equations, similar to the Hartree–Fock equations. However, a price has to be paid for this passage from the wave function to the density as the basic carrier of information. Part of the operator in these one electron equations is unknown: the functional derivative of Exc with respect to ρ(r), δE_xc_ / δρ(r), termed the exchange-correlation potential v_xc_ (r). It can safely be said that the history of DFT is (among others) a quest to find better and better approximations for this unknown v_xc_ (r). This issue will not be the subject of this paper. It is part of the more computational side of DFT, of course, in its evolution, intertwined with a multitude of deep and often subtle concepts, in the end giving rise to the spectacular computational possibilities DFT offers at present as mentioned in the introductory sentences of this text. Needless to say that another decisive factor in all this was the implementation of DFT in nearly all standard quantum-chemical packages and the spectacular and ever-increasing computing power. 

Our interest is devoted to another branch of DFT, conceptual DFT, in which, as will be seen below, in a density-only context, precision is often given to well-known but sometimes rather vaguely defined chemical concepts, such as electronegativity and hardness, thereby affording their numerical evaluation and their use as such or in the context of a number of principles, e.g., the hard and soft acids and bases principle. It turns out that the remaining ingredient in the variational Equation (1), the Lagrange multiplier μ, together with the density itself, are the key ingredients of this endeavor. It all started in 1978 in a landmark paper by Parr and coworkers [8] on the identification of the Lagrange multiplier μ with the Izcowski–Margrave definition of electronegativity, which can be considered as the birth of conceptual DFT. 

## 2. The Basics of Conceptual DFT

In this landmark paper of 1978, a theorem from variational calculus [9] was used to write the left-hand side of Equation (1) as (ꝺE/ꝺN)_v_ so that the Lagrange multiplier in the DFT variational equation proved to be the partial derivative of the system’s energy with respect to the number of electrons at a fixed external potential:μ = (ꝺE/ꝺN)_v_(2)

This expression is highly similar to the electronegativity of an atom in the form presented in the early 1960s by Iczkowski and Margrave [10] when scrutinizing the evolution of atomic energies as a function of the number of electrons for a constant nuclear charge:χ = −(ꝺE/ꝺN)_Z_(3)

Generalizing the condition for constant Z for atoms to a constant v condition for molecules, the Lagrange multiplier can be identified as the opposite of the electronegativity. Going one step further and using a quadratic model for the E = E(N) curve, a finite difference approach then converts χ into Mulliken’s electronegativity definition [11]:χ = ½ (I + A)(4)
where I and A are the first ionization and electron affinity, respectively.

The final result:μ = −χ = −½ (I + A)(5)
shows that the Lagrange multiplier of the Euler Equation (1) has now been identified with a cornerstone of (physical) chemistry: electronegativity, thereby establishing a *bridge between density functional theory and (concepts in) chemistry*. 

Some years later, Pearson and Parr [12] took another important and comparable step by identifying the second derivative of the energy with respect to N at constant v as the chemical hardness η:η = (ꝺ^2^E/ꝺN^2^)_v_(6)
a concept introduced by Pearson in the early 1960s [13] in the context of the study of generalized acid-base reactions. By classifying favorably interacting Lewis acids and bases, he recognized and attributed a predominant role to the polarizability and introduced the qualification “hard” for low polarizable species and “soft” for highly polarizable species. The famous HSAB principle then shows up: hard acids preferentially interact with hard bases; soft acids preferentially interact with soft bases. A key problem that remained was to quantify these new hardness and softness concepts affording, for example, their numerical evaluation. The identification of η as the second derivative (8) provided the missing link for quantitative studies on the hardness of atoms and molecules and to use it as such or in the context of the HSAB principle. This remarkable achievement again linked a chemical concept to DFT, as indeed (ꝺ^2^E/ꝺN^2^)_v_ is nothing else than the N- derivative of the Lagrange multiplier μ. 

It can safely be said that the first phase in the history of conceptual DFT [14] closes in 1984, when Yang and Parr [15] launched the first *local* descriptors, i.e., **r** dependent, and thus varying from place to place. The former descriptors, χ and η, are termed *global* as they are **r** independent and characterize the system as a whole. A mixed second-order derivative:f(**r**) = (ꝺ^2^E/ꝺN δv(**r**)) (7)
was introduced, which is easily seen to boil down to an extension, and generalization of

Fukui’s frontier molecular orbital concept [16]. Indeed, simple perturbation theory [6,17] shows that:(δE/δv(**r**))_N_ = ρ(**r**) (8)
so that f(**r**) can also be written as:f(**r**) = (ꝺρ(**r**)/ꝺN) _v_
(9)
and so, f(**r**) unveils the way a system partitions added or subtracted electrons in space. If the orbitals are kept unchanged (frozen) upon adding or subtracting electrons, it is easily seen that f(**r**) reduces to the HOMO or LUMO density (for decreasing or increasing N, respectively). In this way, a link is established between the function f(**r**) and the basic ingredients of Fukui’s frontier MO theory, highlighting the role of the frontier orbitals in chemical reactions and in describing chemical reactivity. In honor of Fukui, this local descriptor f(**r**) was termed the Fukui function. Note that using Equation (8), f(**r**) can also be written as (δμ/δv(**r**))_N_, again stressing the link with the content of the variational Equation (1). 

In retrospect, a third DFT routed quantity, the functional derivative of the Lagrange multiplier with respect to the external potential, has been connected to a chemical concept, this time describing chemical reactivity. Note that also the electron density itself has entered this series of descriptors as the derivative of E with respect to v(**r**). It is remarkable that three of the main pillars of the fundamental equation of DFT (1), E, v(**r**), and μ, together with the number of electrons N are retrieved as basic ingredients when establishing the link between what could be termed, in view of its origin, the “physicist’s DFT” and what was later termed “the chemist’s DFT”, “chemical DFT”, or most commonly “conceptual DFT”, abbreviated as CDFT. Nowadays, extensive reviews are available on CDFT [17,18,19,20,21,22,23,24] and recently, an authoritative multi-author reference volume was published [25]. 

## 3. Response Functions and Derived Descriptors

The DFT-based atomic or molecular descriptors ρ(**r**), μ, ղ, and f(**r**) share a common feature: they are all functional, partial, or mixed derivatives of the energy with respect to N and/or v. They can be considered as *response functions* characterizing the sensitivity of the system’s energy to perturbations in its number of electrons N and/or its external potential v(**r**). This type of perturbation is quintessential at the onset of a chemical reaction. 

Their role as a response function is easily retrieved when considering the E = E[N, v(**r**)] functional as discussed in Parr and Yang’s book [6], which over the years has fulfilled a decisive role in introducing DFT, and by extension conceptual DFT, in the quantum-chemical community. Changes in N and v of a given species, due to the interaction/reaction with a second species, yield an energy change of the former species, which can be written in a functional Taylor expansion:dE = (ꝺE/ꝺN)_v(r)_ dN + ∫(δE / δv(**r**))_N_ δv(**r**) d**r** = μdN + ∫ρ(**r**) δv(**r**) d**r** + ……(10)
in which, for simplicity, the expansion is stopped at the first order. 

The role of μ and ρ(**r**) as response functions is evident. Extending Equation (10) to higher-order terms then gives rise to a series of response functions associated with higher-order derivatives, either partial, functional, or mixed. In general, they can be written as ꝺ^n^E/ꝺN^m^ δv(**r**_1_) δv (**r**_2_) … δv(**r**_m′_) with n = m + m′. Note that in view of the way the lower-order terms were introduced, in principle, they all bear chemical significance. More precisely, considering the reaction in which a given molecule (reactant 1) is being perturbed by a second one at the onset of the reaction, the coefficients in the expansion (10) clearly depend only on reactant 1 while the details on the approaching reactant are reflected only in the changes dN or δv(**r**), or in reality with a finite perturbation, in ΔN or Δv(**r**). In this way, it becomes clear that these response functions can be considered to represent the *intrinsic* reactivity of the reactant molecule 1 and can be considered as reactivity indicators/descriptors for that molecule. 

With increasing order, the direct use of these reactivity descriptors is expected to be complicated on the one hand due to computational aspects but also due to the increasing intricacies of their interpretation. However, and though a general computational strategy has been put forward by the Ayers group [26], the importance of higher-order terms may be expected to be decreasing when small perturbations are considered as arising at the onset of a reaction, which has been, together with Klopman’s non-crossing rule [27], the philosophy of using these descriptors in comparative studies of reactivity [17,19].

It became commonplace to represent these response functions in a response function tree, which is depicted in Figure 1, until n = 3 [17,20,28,29]. For n = 2, one recognizes the aforementioned chemical hardness as the second pure N-derivative and the Fukui function as the second-order mixed N and v derivative or the derivative of the electron density with respect to the number of electrons at a constant external potential (vide supra). Due to the discontinuity of the electron density with the number of electrons [6], it is customary to define a Fukui function both on the electron-deficient and -abundant side of the considered integer N, written as f^−^(**r**) and f^+^(**r**). In a finite difference approach, one then obtains:f^−^(**r**) = ρ_N_ (**r**) − ρ_N−1_ (**r**) and f^+^(**r**) = ρ_N+1_(**r**) − ρ_N_(**r**) (11)

In practice, often atom-condensed Fukui functions are used either by the use of a population analysis technique (originally Mulliken charges [30] were chosen) or by numerical integration of Equation (11) over the volume attributed to a given atom k. One then obtains the following working equations [31] applicable for an electrophilic, nucleophilic, and radical attack on the considered reactant:f_k_^−^ = q_k_ (N) − q_k_ (N − 1) (12a)
f_k_^+^ = q_k_ (N +1) − q_k_ (N)(12b)
and f_k_^0^ = ½ (q_k_ (N +1) − q_k_ (N − 1))(12c)

The final equation, used in case of a radical attack, results from an averaging of the equations for electrophilic and nucleophilic attack. Similar expressions extending the Fukui function and other CDFT quantities from atoms to functional groups properties have been put forward by De Proft and Geerlings [32]. 

The remaining second-order response function is (δ^2^E/δv(**r**) δv(**r**′))_N_, termed the linear response function χ(**r**, **r′**), [6] as it represents the linear term in the response of the density ρ(**r**) to a perturbation v at a point **r′**:χ(**r**, **r′**) = (δ^2^E /δv(**r**) δ(v(**r′**))_N_ = (δρ(**r**)/δv(**r′**) _N_
(13)

It has been scrutinized only in the last 15 years, mainly by Geerlings and coworkers, addressing both its computability, interpretation, and especially its chemical relevance, e.g., in the context of inductive and mesomeric effects, aromaticity, transferability of functional groups, and molecular conductivity [33,34,35,36].

The hyper-hardness (ꝺ^3^E/ꝺN^3^)_v_, the N- derivative of the hardness, denoted as η^(2)^ or γ, and the simplest third-order response function, was introduced by Parr and Fuentealba in 1991 [37]. It turned out to be of limited chemical significance and thereby received considerably less attention than the hardness itself. The most rewarding n = 3 response function from a chemical point of view turned out to be the dual descriptor f^(2)^(**r**) introduced by Morell and Grand in 2005 [38,39] as the N- derivative of the Fukui function:f^(2)^ (**r**) = (∂f(**r**)/∂N)_v_
(14)

As a particularly attractive property, this descriptor was shown to provide a one-shot picture of both the electrophilic and nucleophilic regions in a molecule, promoting it to an excellent tool for applying and scrutinizing the HSAB principle at the local level (vide infra). The remaining n = 3 derivatives (the Fukui response function and the quadratic density response function (see Figure 1) and the n = 4 derivatives largely remain unexplored hitherto. 

Though fundamental, as energy response functions, the above-mentioned descriptors are not the only acceptable descriptors. Without going into the technical/mathematical details too much, it should be stressed that a new series of descriptors appear, among which the widely used global and local softness, when writing the perturbation expansion not for the E = E [N,v] functional but for the Ω = Ω [μ, v] functional, i.e., when passing, via a Legendre transformation, from the so-called canonical ensemble to the grand canonical ensemble [6,40]. The latter approach is better suited for open systems in which the number of electrons is replaced by the chemical potential as the fundamental variable. A second response function tree analogous to the one depicted in Figure 1 can then be set up (see, for example, [17]), in which, besides N (in fact its opposite) and again ρ(**r**) as first-order derivatives, the global softness S, the local softness s(**r**) [41], and the softness kernel s(**r**,**r**′) [42] appear as counterparts of the global hardness, the Fukui function, and the linear response function. 

The local softness definition in the grand canonical ensemble:s(**r**) = (∂ρ(**r**)/∂μ)_v_(15)
can easily be written as a simple product of the global softness (the inverse of the global hardness) and the Fukui function: s (**r**) = (∂ρ(**r**)/∂N)_v_ (∂N/∂μ)_v_ = f(**r**) (1/η) = S f(**r**) (16)
showing that the Fukui function redistributes the global softness of a system among different regions. It gained widespread use mostly also in an atom-condensed version analogous to the condensed Fukui function Equation (12) as:s_k_
^−,+,o^ = S f_k_ ^−,+,o^(17)

In addition to the evident usefulness of the *global* softness, it has been recognized quite early in the CDFT literature that the *local* softness is more suitable than the Fukui function when comparing, e.g., reactivity along a series of molecule, whereas the Fukui function itself is sufficient to compare, for example, the relative reactivity of different sites within a given molecule (intermolecular vs. intramolecular reactivity sequences). Similar considerations apply to the grand canonical ensemble counterpart of the linear response kernel, the softness kernel s(**r**,**r′**). 

Finally, one word of comment on the counterpart of the local softness, the local hardness η(**r**). Several expressions/working equations have been proposed after its introduction by Ghosh and Berkowitz [42], with one of the simplest being [43]:η(**r**) = (δμ/δρ(**r**)_v_
(18)
ensuring that the product of local softness and hardness integrates to 1:∫s(**r**) η(**r**) d**r** = 1 (19)

Due to an ambiguity in the functional derivative [44] (the first Hohenberg Kohn theorem highlights the dependence between ρ(**r**) and v(**r**)), it has, however, been recognized that this definition is problematic. Without going into details (for critical accounts, see, for example, [45,46]), the overall results with different (though sometimes equivalent) expressions presented until now should be considered with caution as the problem has not been fully settled yet. 

Another way to broaden the number of descriptors and adapt them to a larger variety of reaction conditions is to increase the number of variables in the E = E[N,v] functional. The first and in a quantum-mechanical context very natural extension was to include spin variables. In the late 1980s and early 1990s, Galvan, Gazquez and Vela [47], and Ghanty and Ghosh [48], therefore, considered the functionals E = E [N, N_S_, v, **B**] and E = E [N_α_, N_β_, v_α_, v_β_ ], respectively. These two intimately related approaches differ in the sense that in the former approach, spin polarization is included whereas in the latter, the number of electrons is resolved into its spin components. N_S_, the spin number, represents the difference between the number of α and β electrons (N_α_ and N_β_). This approach paved the way to study the reactivity of atoms and molecules under perturbation of their spin state typically occurring by a magnetic field **B** or by spin transfer from its environment or another reagent. 

A more recent extension has been the inclusion of temperature by the Ayers group, of fundamental importance when trying to cope with the N-differentiability problem of the E = E[N,v] functional [49]. Perdew et al. indeed showed that at constant v, its N dependence is a series of straight lines intersecting at integer N in the zero-temperature limit [50]. This, however, leads to problematic expressions for the hardness (and by extension all other second and higher N-derivatives) as it becomes the derivative of a step function. This problem has been coped with by the Ayers group’s introduction of temperature in the description of the system, thereby ensuring differentiability of the E = E (N) function. An open system ensemble average electronic energy (and its derivatives) then turns out to be the central quantity in this finite temperature chemical reactivity theory [50,51]. For the practicing chemist, however, it should be noticed that the temperature values at which the deviation from the zero-temperature limit becomes meaningful by far exceeds the usual laboratory conditions so that, e.g., in the applications at stake in the present Special Issue, one can safely stick to the zero-temperature approach. 

A recent series of extensions was the introduction of external electric and magnetic fields in the energy functional. Pioneered by Chattaraj in 2003 and 2014, respectively [52,53], they were not addressed frequently until now. In recent years, however, De Proft, Geerlings, and coworkers presented detailed studies on the influence of these external fields on a variety of reactivity descriptors [54]. The influence of oriented external electric fields (OEEFs) was studied both for global and local descriptors, leading in the latter case to considerations on electric-field-induced asymmetry of local descriptors (e.g., the Fukui function) with concomitant consequences for the selectivity in particular conditions [55]. In a comprehensive study on the influence of an external magnetic field on the atomic electronegativity and hardness of the main group elements of the periodic table, the change in the electronic configuration upon increasing the field strength turned out to be of fundamental importance when scrutinizing periodicity [56]. This group also considered the inclusion of an external mechanical force [57,58], fundamentally different both in its nature and in its computational approach to the electromagnetic fields, and very recently external pressure [59], as a natural extension to earlier work on the influence of confinement [60,61]. Both extensions fit into recent evolutions in experimental chemistry in the field of mechanochemistry and high-pressure chemistry, respectively [62,63].

Finally, returning to idea of response functions, be it in the canonical or grand canonical ensemble, it should be stressed that besides these response functions, other types of descriptors are also acceptable with the proviso that they have a firm physical basis and are constructed with mathematical rigor. In a recent status paper, it has been stressed that, for example, other descriptors derived from the E = E[N,v] functional and exploitation of its characteristics are perfectly acceptable.

Parr’s electrophilicity ω [64] is the most prominent example and has played a predominant role in applications of CDFT to bio-active compounds ([65,66] (vide infra)). It refers to the position of the minimum in the quadratic interpolation for the E(N) curve at constant external potential and yields the energy gain at the system’s maximal uptake of electrons from an electron reservoir. A simple expression then follows:ω = μ^2^/ η (20)
where two response functions μ and η are combined. This type of descriptor is termed a “derived” descriptor and as seen in Equation (20); they nearly always boil down to products/ratios of the previously discussed response functions. Just as in the case of the response function, this global descriptor can be made local by multiplying it by a function distributing this property in space, which, for example, the Fukui function does with the softness in Equation (16). One then arrives, in the case of an electron uptake by an electrophile, to a local electrophilicity index [67,68]:ω^+^ (**r**) = ω f^+^ (**r**) (21)
and analogous expressions when involving f^0^ (**r**) and f^−^ (**r**). In an atom-condensed form, one then arrives at the so-called philicity indices [67]:ω_k_ ^−,+,o^ = ω f_k_
^−,+,o^
(22)

Using the dual descriptor f^(2)^(**r**) to partition the electrophilicity over the different atomic regions, one obtains, again in an atom-condensed form, the so-called multiphilic descriptor for a given atom k as [69]:Δω_k_ = ω f_k_^(2)^
(23)

Finally note that in the above-mentioned status paper, it is explicitly mentioned that one should avoid combining reactivity descriptors (response functions) in an ad hoc fashion without conferring them any physical or chemical meaning as opposed to the case of electrophilicity.

## 4. Principles 

The literature on conceptual DFT shows that in the majority of applied papers, the above discussed concepts (response functions and derived descriptors) were either used “as such” or, and to a broad extent, in the context of “principles”, which can be characterized as “rules of thumb” to interpret/predict the direction of a reaction, sometimes concentrating on its kinetic aspects, sometimes on its thermodynamics. The three traditional principles are: the electronegativity equalization principle, the hard and soft acids and bases principle, and the maximum hardness principle. We briefly outline these principles, in chronological appearance in the literature, paying, however, somewhat more attention to the HSAB principle, which most probably will turn out to be most frequently addressed in the applications that are at stake in this Special Issue. 

The *electronegativity equalization principle* is to some extent an outlier when considering intermolecular interactions or reactions as in this Special Issue. Indeed, it essentially concentrates on the charge distribution of a given species. Formulated already in 1951 by Sanderson [70], long before the advent of DFT, let it be CDFT, it postulates that upon molecule formation, the electronegativities of all constituent atoms equalize, yielding a molecular electronegativity equal to the geometrical mean of the original atomic electronegativities [71]. In 1978, Donnelly and Parr [72], shortly after the landmark paper on the identification of the electronic chemical potential as electronegativity (cf.§2), proved the constancy of the electronic chemical potential over the considered system. Theoretical and numerical evidence for the geometrical mean postulate was presented by Parr and Bartolotti in 1982 [73]. Sanderson’s postulate was thereby given a sound theoretical basis. The step from electronegativity equalization towards an electronegativity equalization method (EEM) was taken by Mortier and coworkers around 1985 [74,75], turning the principle into an easy-to-implement computational ansatz capable of calculating the charge distributions in polyatomic molecules. Nowadays, EEM can be used and is used (e.g., via its implementation in widely used molecular mechanics/ force field packages) to yield a reasonable first estimate of the charge distributions in large series of (not too exotic) large molecules, for example, of importance in biomolecular systems (as in drug discovery research), be it, as should be repeated, that interactions as such are not the objective of the ansatz [76]. 

The *hard and soft acids and bases* principle was already touched upon in §2 describing how the identification of the second derivative of the energy with respect to the number of electrons as the chemical hardness was achieved in the context of Pearson’s hard and soft acids and bases principle. After the introduction of hardness and softness in CDFT, a formal proof for the HSAB principle was given in 1991 by Chattaraj, Lee, and Parr [77] for its application at a global, i.e., molecular, level. Mendez and Gázquez incorporated the HSAB principle at a local level, thereby focusing on the interaction characteristics between the relevant atoms of the interacting acid and base [78]. Note that in applications at the global level, mostly stability issues (thermodynamic in nature) are at stake, whereas in studies at the local level, often discussing regioselectivity problems, the reactivity aspect (kinetic in nature) is predominant. In general, it can be said that the HSAB principle has found a firm place in the CDFT community (and even in a much broader “general chemistry” context) and that the (applications of) HSAB principle do form a substantial part of the CDFT literature. Many successful applications were reported but also some (most probably underreported [19]) failures. The basic reason for this can be found when scrutinizing Pearson’s words when he formulated the principle, stating that “all other things being equal, hard acids prefer binding to hard bases and soft acids to soft bases” [79]. The “all other things being equal” caveat is indeed often forgotten when discussing results (both positive and negative ones): although never perfectly satisfied, the conditions under which the HSAB principle are applied should always be scrutinized. It is, therefore, not surprising that in the above-mentioned status paper, a section was devoted to the domain of validity of (among others) the HSAB principle and that it is stated that the CDFT community “should put further effort in exploring the domains where the HSAB principle holds and establish the caveats that must be remembered when applying the HSAB principle both at the global (stability) and local (reactivity) levels”.

Pearson’s *maximum hardness principle* (MHP), dating from the late 1980s [80], is the third principle in the context of CDFT. Pearson’s original statement that “there seems to be a rule of nature that molecules arrange themselves to be as hard as possible” was turned into a more formal CDFT framework by Parr and Chattaraj in a proof they presented in 1991 [81]. Although this paper has been widely cited, MHP has found less acceptance outside the CDFT community than the HSAB principle and a much smaller number of applied literature, also on biosystems, invoked MHP. A possible reason is that the constraints under which MHP is rigorously valid (e.g., constant chemical and external potential) are highly restrictive, leading to an even more important caveat in its application than in the case of the HSAB principle. Further theoretical support and a detailed analysis of the conditions to be fulfilled for thoughtful application of MHP have very been recently put forward by Miranda-Quintana et al. [82]. 

On the other hand, in view of the widespread use of the electrophilicity in, among others, studies on biosystems, a more recent fourth principle, Chattaraj’s *minimum electrophilicity principle* (2003), should be mentioned for which recent detailed studies showed encouraging statistical performance data as compared to MHP [83]. 

## 5. Applications

A recent educated guess on the volume of the CDFT literature led to an order of magnitude of 4000 papers [19]. CDFT, as a subfield of DFT, has clearly been the subject of intense intellectual activity in the past decades. Whereas, after its launching in 1978, the number of papers remained relatively small in the 1980s, with most papers being fundamental in nature, a steady increase manifested itself in the 1990s with a balance between fundamental and applied studies. In the early years, these applications mainly concentrated on reactivity studies in classical “textbook” organic reactions scrutinizing different types of organic reaction types/mechanisms, (general) acid-base and complexation reactions, and an already important series of studies on clusters and catalysis. This clearly shows up in the 2003 review by Geerlings, De Proft, and Langenaeker [17], presenting an almost complete literature survey on both fundamental and applied CDFT until that time. After 2000, an avalanche of papers were published where applications clearly dominate and which were written not only by theoreticians but also, and more and more, by experimentalists, often in collaboration with theoreticians. 

It is natural that, in view of the complexity of biological molecules or more generally interactions with biosystems, applications of CDFT to these systems started to appear in that period together with a hard-to-describe extension of the field of applications. Nowadays, applications are published across nearly all branches of chemistry, from inorganic and materials chemistry to organic, organometallic, and polymer chemistry as just mentioned on biosystems. Concentrating on the latter subject, it is not the intention to give in this introductory paper of this Special Issue the complete bibliography of this subfield; the reader will obtain an idea of it when reading the complete Special Issue, but with the danger of omitting important contributions in this personal account, we will try to situate a few developments/research lines in the last twenty years. Indeed, before the change of century, not that many papers applying CDFT to biomolecules appeared, as witnessed by their absence in the aforementioned 2003 Chemical Reviews. 

In 2004, an important paper appeared by Khandogin and York [84] introducing, besides 3D plots of the molecular electrostatic potential [85], at that time already well-known/used in the quantum-biochemical community [86], similar plots for the local hardness (pointing out its intricacies, vide supra) and a discussion of the Fukui indices on several biologically important peptides in solution, also including a continuum solvent model. In the context of the HSAB principle, Rivas et al. evaluated in 2004 the group softness to locate and orientate reactive regions in the hydride transfer reaction between the isoalloxazine moiety of flavins and the nicotinamide moiety of NAD(P)H as involved in flavoenzyme catalysis [87]. The electrophilicity turned out to be a successful descriptor of the overall reactivity pattern of these systems. Around the same time, Roos et al. started a series of investigations on the use of CDFT descriptors, global ones such as the softness and electrophilicity and local ones such as the Fukui functions, in investigating *enzymatic catalysis* [88,89,90,91,92] (a heading under which the two previous contributions can also be placed) for a diversity of systems, with particular attention to the application of the local HSAB principle [93]. They also offered an outlook to the future foreshadowing an extension of their “gas-phase” models by implementing larger parts of the enzyme (QM/MM) on the one hand, and by linking structure and dynamics via molecular dynamics (MD) on the other hand, in view of the ever-increasing computing power in accompanying methodological advances. Some years later, Faver and Merz treated systems of increasing size in investigations on ligand docking, active site detection, and even protein folding, again with a predominant role for the Fukui function in the context of the HSAB principle [93]. In recent years, the work by Grillo et al. should be mentioned, who, using semi-empirical Hamiltonians, obtained reactivity descriptors for large biological systems with reasonable accuracy and speed. They took one step further, simulating reaction paths involving three enzymatic systems (triosephosphate isomerase, haloalkane dehalogenase, and adenosisne kinase), thereby following the evolution of, among others, the local hardness via a new working equation [94,95]. Another step further was the use of a Boltzmann weighted atom-condensed Fukui function by Oller, Saez, and Vöhringer-Martinez [96], thus taking into account conformational fluctuations in a study on the enzymatic fixation of carbon dioxide.

A fundamentally different research line was launched by Chattaraj and coworkers, at about the same time as the CDFT/enzymatic catalysis line sketched above, on the use of CDFT descriptors to quantify the *biological activity and/or toxicity* of a variety of mainly organic molecules, e.g., different groups of polyaromatic hydrocarbons (PAH) [97,98,99]. Electrophilicity and/or philicity, both in its global and local form, largely explored by this group for years (vide supra [65,66,67]), were used as a key CDFT reactivity descriptor and injected in a QSAR approach to explain the trends in, for example, the pCI50 values as indicators of biological activity.

We end this far-from-complete overview with mention of a recent research line by Glossmann-Mitnik and coworkers, again different from the previous ones, investigating the chemical and biological reactivity of certain groups of peptides combining CDFT descriptors with tools from cheminformatics. Studies on marine cyclopeptides involved most of the global CDFT descriptors, combined with the Fukui function and the dual descriptor as local ones, scrutinized these structures for their potential therapeutic abilities. In a more general context, they inject CDFT into “computational peptidology” [100,101,102,103].

The evolution sketched above should be accompanied by a parallel endeavor of CDFT specialists to provide well-documented, user-friendly software packages, which can easily be coupled to standard quantum-chemical programs [19]. The ChemTools package, designed to support arbitrary energy models and reactivity indicators of arbitrarily high-order [26,104] and the already widely used Multiwfn package offering the evaluation of a variety of CDFT descriptors [105] have been important steps in this direction.

## 6. Conclusions

Conceptual density functional theory offers a broad spectrum of tools for studying a variety of properties of an atom or molecule relevant upon its interaction with another system. Within a perturbational approach and starting from the E = E [N,v] functional (or its grand canonical ensemble counterpart), a series of response functions appear in a natural way, sharing a common solid physical basis intertwined with mathematical rigor, describing the intrinsic reactivity of a molecule upon perturbation by an interacting partner, external electric and magnetic fields, mechanical forces, confinement, or external pressure. Electronegativity, hardness, softness as global descriptors, the electron density itself, the Fukui function, the local softness and the dual descriptor as local descriptors, and the linear response function and its counterpart the softness kernel as non-local descriptors are the most prominent members of these descriptors. “Derived descriptors” such as the electrophilicity of “generalized” philicity, derived from the properties of the E = E (N) function, complete the picture. Used as such or in the context of principles such as the electronegativity equalization principle, the HSAB, the maximum hardness, and minimum electrophilicity principle, these descriptors found widespread use, leading to an avalanche of papers on applications covering a large variety of subfields of chemistry and neighboring sciences. Thanks to methodological evolutions but also through the impressive evolution in software and hardware, biosystems of ever-increasing size gradually entered this application portfolio in the past twenty years, with studies in fields (among others) varying from enzymatic catalysis, via biological activity and /or toxicity of organic molecules to computational peptidology. On the basis of this ongoing evolution, one can expect that “the best is yet to come”.

## Figures and Tables

**Figure 1 pharmaceuticals-15-01112-f001:**
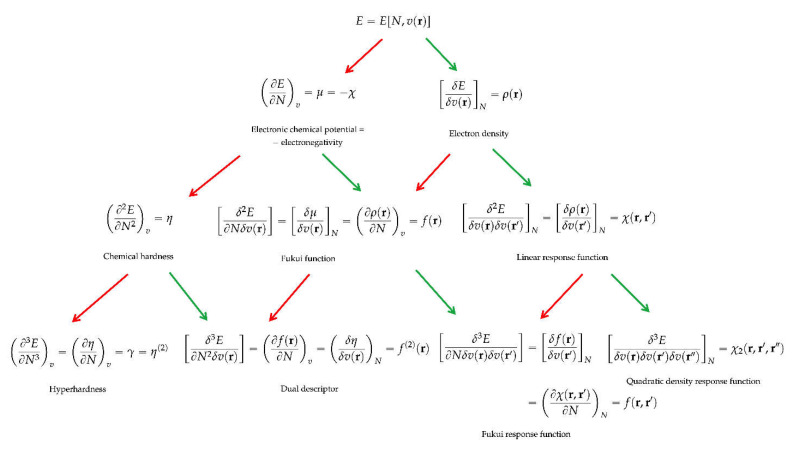
Response function tree of the energy E vs. changes in the number of electrons N and the external potential v(**r**). Red arrows indicate differentiation with respect to N, and green arrows indicate differentiation with respect to v(**r**). Reprinted with permission, from *Conceptual Density Functional Theory*, S. Liu Editor, 2022, Wiley VCH, Weinheim, Germany, Chapter 2: F. De Proft “Basic Functions” [29].

## Data Availability

Data is contained within the article.

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
