# Peer review of "From Density Functional Theory to Conceptual Density Functional Theory and Biosystems"

_pharmaceuticals, 2022, doi:10.3390/ph15091112_

Round 1
Reviewer 1 Report
This review paper is an interesting contribution to the discussion on the application of the conceptual density functional theory. The use of CDFT has been steadily expanding across a wide range of chemistry-related specializations, including organic and inorganic chemistry, polymer and materials chemistry, catalysis, and nanotechnology, among others. The ever-increasing scale of the systems under investigation has been successfully managed, not only as a result of developments in research methodology but also as a result of significant advances in computer software and hardware. Can the author mention some of these software and hardwares in the introduction section and the discussion.
There are some errors in spelling, syntax, punctuation, usage of capital letters, and consistency in language style in the manuscript.
Reviewer 2 Report
The present contribution from Paul Geerlings, titled "From density functional theory to conceptual density functional theory and biosystems', highlights the evolution of conceptual density functional theory and touches upon the applications of various DFT descriptors such as electronegativity, hardness, softness, Fukui function, linear response function, etc. This is yet another impressive scientific highlight of conceptual density functional theory. The article in the present form fits well to the very high standards and justifies publication in the journal 'Pharmaceuticals'. I recommend the publication of this manuscript in Pharmaceuticals in its present form.
Author Response
"Please see the attachment."
